# Making a Decision between Acute Appendicitis and Acute Gastroenteritis

**DOI:** 10.3390/children7100176

**Published:** 2020-10-11

**Authors:** Yi-Ting Lu, Po-Cheng Chen, Ying-Hsien Huang, Fu-Chen Huang

**Affiliations:** 1Department of Pediatrics, Kaohsiung Chang Gung Memorial Hospital, Kaohsiung 833, Taiwan; b9702093@cgmh.org.tw (Y.-T.L.); yhhuang@cgmh.org.tw (Y.-H.H.); 2Department of Physical Medicine and Rehabilitation, Kaohsiung Chang Gung Memorial Hospital, Kaohsiung 833, Taiwan; b9302081@cgmh.org.tw; 3Department of Public Health, College of Medicine, National Cheng Kung University, Tainan 701, Taiwan; 4Department of Pediatrics, College of Medicine, Chang Gung University, Taoyuan 333, Taiwan

**Keywords:** acute appendicitis, gastroenteritis, abdominal pain, diarrhea

## Abstract

Acute appendicitis is one of the most common pediatric abdominal emergencies. Early diagnosis is vital for a positive outcome. However, it may initially present with diarrhea and vomiting, mimicking acute gastroenteritis, thus delaying prompt surgery. Differentiating appendicitis from gastroenteritis in a timely manner poses a challenge. Therefore, we aim to investigate the predictors that help distinguish acute appendicitis from acute gastroenteritis. We conducted a retrospective case-control study, evaluating children admitted due to abdominal pain with diarrhea. Subjects were divided into two groups according to the final diagnoses: acute appendicitis and acute gastroenteritis. We adopted multiple logistic regression analysis and the area under the receiver operating characteristic curve to identify independent predictors of acute appendicitis and select the best model. A total of 32 patients diagnosed with appendicitis and 82 patients with gastroenteritis were enrolled. Five independent predictors of acute appendicitis included vomiting, right lower quadrant (RLQ) pain, stool occult blood (OB), white blood cell (WBC) count, and C-reactive protein (CRP). The revised combined model exhibited a higher degree of discrimination and outperformed the pediatric appendicitis score (PAS) model. In conclusion, our study was proved to be helpful for assessing cases with abdominal pain and diarrhea in order to more accurately distinguish appendicitis from gastroenteritis in children in a timely manner.

## 1. Introduction

Acute appendicitis in children is the most common acute surgical condition but remains a difficult diagnosis for clinicians. In children, it has a higher perforation rate than in adults and may present with complicated diseases in up to 40% of cases [1]. Early diagnosis is vital to improve outcomes and avoid complications such as appendiceal perforation, abscess formation, and postoperative complications.

Pediatric appendicitis score (PAS) has generally been utilized to diagnose appendicitis in children with abdominal pain [2]. However, many cases receive a score that signifies intermediate risk (a PAS score of 4–6), encouraging clinicians to seek surgical consultation or advanced diagnostic imaging [3]. 

In clinical practice, acute appendicitis sometimes mimics acute gastroenteritis. Since enteric infections can cause appendicitis, diarrhea may be an initial manifestation, which makes gastroenteritis the assumed diagnosis [4,5,6,7]. The major factor in the delayed diagnosis of acute appendicitis is suspected gastroenteritis, which thus delays timely surgery [8]. However, differentiating acute appendicitis and acute gastroenteritis in the early stage still poses a challenge for clinicians. 

This study investigates clinical characteristics, laboratory data, and image studies between cases with acute appendicitis mimicking gastroenteritis and those with gastroenteritis mimicking appendicitis. The purpose of this study is to identify the clinical predictors that may assist physicians in distinguishing acute appendicitis from acute gastroenteritis in children. 

## 2. Materials and Methods

### 2.1. Study Population

A retrospective case-control study was conducted in the Department of Pediatrics, Kaohsiung Chang Gung Memorial Hospital (KCGMH), Kaohsiung, Taiwan from 1 January 2015 to 30 April 2020. KCGMH is a teaching medical center in southern Taiwan that provides primary to tertiary care for children younger than 18 years of age. Taiwan National Health Insurance (NHI) covers more than 99% of Taiwan citizens and pays out every prescribing medicine, clinic visit, admission, and surgery.

### 2.2. Enrolment Criteria

The enrolment criteria were as follows: children <18 years old, who were admitted due to abdominal pain with diarrhea and/or vomiting, and final diagnoses were acute appendicitis or acute gastroenteritis. Other diagnoses were excluded. Subjects were divided into two groups according to their final diagnosis: acute appendicitis (group 1) and acute gastroenteritis (group 0). Acute appendicitis was diagnosed according to clinical presentation, radiographic study, surgical report, and pathology. Patients came from the emergency department and ordinary wards in KCGMH.

### 2.3. Clinical Covariates

Age and sex were confirmed by the NHI database, and baseline characteristics were collected. Data were retrieved from validated hospital discharge, outpatient visit, emergency department, and laboratory documents. These include diagnoses of acute gastroenteritis, infectious colitis, acute appendicitis, and ruptured appendix. We gathered data related to clinical symptoms and signs, physical examination, laboratory data, abdominal sonography, abdominal computed tomography (CT) reports, and operation records. Furthermore, we evaluated potential parameters for differentiating acute appendicitis from gastroenteritis. Potential clinical predictors included (1) clinical symptoms and signs, such as the duration of fever, abdominal pain and diarrhea, presence of vomiting, diarrhea frequency, the pattern and location of abdominal pain, and the characteristic of bowel sounds; (2) laboratory findings, such as leukocyte count, differential count (segment %, lymphocyte %, and eosinophil %), hemoglobin level, platelet count (PLT), C-reactive protein (CRP), aspartate aminotransferase (AST), alanine aminotransferase (ALT), presence of occult blood (OB) or pus in the stool, and stool culture; and (3) image findings, such as abdominal sonography and abdominal CT.

### 2.4. Statistical Analysis

In descriptive statistics, demographic characteristics were presented as either mean ± standard deviation for continuous variables or count numbers for categorical variables. Differences in variables between the appendicitis group and the gastroenteritis group were compared using independent Mann–Whitney U test for continuous variables and chi-square test or Fisher’s exact test for categorical variables. We adopted the Benjamini–Hochberg method [9] to control the false discovery rate of multiple testing in our study. For statistically different variables between two groups, we performed multivariable stepwise logistic regression for clinical variables and laboratory variables to separately develop a clinical predictive model and a laboratory predictive model. Finally, we combined the clinical and laboratory predictive models to create a combined predictive model using multivariable stepwise logistic regression. The criteria used in the model selection methods for stepwise regression applied the Akaike’s information criteria [10,11,12]. Receiver operating characteristic (ROC) analysis was used to calculate the area under the curve (AUC) of each significant predictor in the combined predictive model. After the initial combined model was obtained by the stepwise logistic regression, we gave each predictor an assigned point value to calculate the revised combined model score. Furthermore, we compared the AUC of the combined predictive model with that of the predictive model using PAS. A two-tailed p-value less than 0.05 was considered statistically significant. Statistical analyses were performed in SAS software, Version 9.4 of the SAS System for Windows (SAS Institute Inc., SAS Campus Drive, Cary, North Carolina 27513, USA).

### 2.5. Ethics Statement

Ethical approval for this study was granted by the Institutional Review Board of Chang Gung Memorial Hospital (Kaohsiung, Taiwan) (No. 202000311B0). We retrieved patient lists from the electronic database and retrospectively reviewed medical information from medical records. The need for consent was waived due to the retrospective nature of the project and the anonymous analysis of data.

## 3. Result

### 3.1. Demographics and Clinical Characteristics

During the study period, 114 cases (63% male) were identified. Thirty-two cases were diagnosed with appendicitis (group 1), and 82 cases were diagnosed with gastroenteritis (group 0). In group 0, the mean ± standard deviation of age was 10.47 ± 3.85 years. In group 1, the mean ± standard deviation of the age was 8.88 ± 4.51 years.

The demographics and clinical characteristics of the two groups are shown in Table 1. Among the categorical characteristics, vomiting, more than three days of abdominal pain, right lower quadrant (RLQ) pain, hypoactive bowel sound, presence of the peritoneal sign, and absence of stool occult blood (OB) were significantly observed in group 1 (*p* = 0.01, <0.001, 0.008, 0.002, 0.004, and 0.004, respectively). Among continuous characteristics, hospitalization days, PAS, white blood cell (WBC) count, platelet count, absolute neutrophil count (ANC), and CRP level were significantly higher in group 1 (*p* <0.0001, <0.0001, <0.0001, 0.0125, <0.0001, and <0.0001, respectively), as shown in Figure 1 and Appendix A.

The stool analysis in Table 2 indicates that positive stool OB was significantly detected in group 0 (*p* = 0.004), while positive stool pus shows no significance in two groups. 

In patients with gastroenteritis, the stool culture of 15 (18%) patients yielded Campylobacter jejuni, six patients (7%) yielded *Salmonella* species, five patients (6%) yielded Rotavirus, and one patient (1%) yielded Norovirus. The rate of positive stool cultures was 34%. Negative stool culture was recorded in all patients of the appendicitis group. 

### 3.2. Multivariable Stepwise Logistic Regression of Predictors for Acute Appendicitis

Five independent predictors of acute appendicitis, which we identified through multivariate analysis logistic regression, were vomiting (odds ratio [OR], 6.69; 95% confidence interval [CI], 1.37–32.72; *p* = 0.019); RLQ pain (OR, 9.06; 95% [CI], 2.06–39.80; *p* = 0.004); absence of stool occult blood (OB)(OR, 0.05; 95% [CI], 0.00–0.73; *p* = 0.028); WBC count (OR, 1.2; 95% [CI], 1.07–1.34; *p* = 0.002); and CRP level (OR, 1.19; 95% [CI], 1.19–1.30; *p* < 0.001), as shown in Table 3.

The areas under the curve (AUC) for vomiting, RLQ pain, negative stool OB, higher WBC count, and CRP level were 0.63, 0.65, 0.62, 0.79, and 0.79, respectively. Table 4. indicated different variables and initial combined variables ROC models associated with acute appendicitis. Figure 2 shows the ROC curves of different variables and the initial combined five-variable model for differentiating between appendicitis and gastroenteritis.

Table 5 and Table 6 show that our revised combined model compromises these five variables with assigned point value to calculate the score. In our revised model, vomiting and RLQ pain were assigned a score of 2; WBC > 18.2(10^3^/uL) and CRP > 7.64(mg/dL) were assigned a score of 1; positive stool OB was assigned a score of −3. The cut-off point was 3, as shown in Table 5. The sensitivity of the revised model was 89%, and the specificity was 78%. The cut-off point of PAS was 4 based on our enrolled patient’s data. 

The revised combined model of these five predictors (AUC = 0.90) for predicting appendicitis was more significant than the pediatric appendicitis score (PAS) model (AUC= 0.80), *p* = 0.012, as shown in Table 5 and Figure 3.

### 3.3. Abdominal Sonography and CT Scan

All of our cases received abdominal sonography during admission. The sensitivity of abdominal sonography was 84%, and the specificity was 95%. The positive predictive value (PPV) was 87%, and the negative predictive value (NPV) was 94%. Twenty-five patients (78%) received abdominal CT in the appendicitis group, and 18 patients (22%) received abdominal CT in the gastroenteritis group. The sensitivity of CT was 96%, and the specificity was 100%.

### 3.4. Appendiceal Perforation

In our study, the rate of appendiceal perforation with abscess formation at the time of hospital evaluation was 42%.

## 4. Discussion

Appendicitis in children has a broad spectrum of clinical presentation, with signs and symptoms varying greatly. Prior studies have indicated that the detection of appendicitis in children is often delayed due to misdiagnosis. Reported rates of misdiagnosis range from 7.5% to 37% in children [4,13,14]. Many studies have reported that enterocolitis is the most common diagnosis in cases of misdiagnosed appendicitis [5,14,15,16]. A history of diarrhea is an important factor that complicates the diagnosis, prolongs the observation period, and delays appropriate therapy [4,6]. 

Sonography is a useful method for the early assessment of acute appendicitis [1,7,14]. In our study, the sensitivity and specificity of abdominal sonography for appendicitis was 84% and 95%, respectively. Despite good sensitivity and specificity, sonography is an operator-dependent technique and is not available in all medical facilities. CT scan has been a gold standard imaging study for evaluating suspected appendicitis [1]. In our study, the sensitivity of a CT scan for appendicitis was 96%, while the specificity was 100%, but a CT scan results in radiation exposure and increases costs. Reductions in CT scans for appendicitis have been observed in a national sample of 35 pediatric institutions. Nevertheless, rates for appendiceal perforation have remained unchanged [17]. The rate of appendiceal perforation in our study was as high as 42%. Therefore, we need other practical predictors for clinical assessment. 

In clinical presentations, vomiting and RLQ pain were independently significant in our study. Vomiting is a common feature of gastrointestinal upset but is not specific to acute gastritis [18]. However, profuse vomiting may indicate ileus and the development of generalized peritonitis after perforation [19]. On the other hand, previous studies indicated that the presence of RLQ tenderness is probably the most sensitive physical finding in early appendicitis [20]. In PAS, RLQ pain is also a single diagnostic variable with a score of 2 (one of the highest-score factors in the score system). Since appendicitis has a broad spectrum of clinical presentation, we then integrated five predictors (vomiting, RLQ pain, negative stool OB, higher CRP level, and higher WBC level) into a revised combined model. This revised combined model exhibited a high degree of discriminating acute appendicitis among patients with abdominal pain and symptoms of gastroenteritis. Furthermore, it outperformed the PAS model.

PAS is a weighted clinical scoring system with eight clinical features for accessing abdominal pain and diagnosing appendicitis in pediatric patients [2]. This score combines history, physical, and laboratory data to assist in the diagnosis. Migration of pain, anorexia, vomiting, fever >38 °C, leukocytosis and polymorphonuclear neutrophilia were each assigned a score of 1; RLQ pain and cough/percussion/hopping tenderness were each assigned a score of 2. A total score of >6 may be compatible with the diagnosis of appendicitis. However, the patients we enrolled in the study all suffered from abdominal pain and symptoms mimicking gastroenteritis, resulting in a different cut-off score of PAS (4) from previous studies. 

In laboratory examinations, the absence of stool OB is an independent predictor for appendicitis in this study. Diarrhea accompanying appendicitis is usually culture-negative and is limited to the release of small amounts of loose stool without blood or mucus rather than the copious amounts of stool in enteritis [21]. In contrast, bacterial colitis often results in inflammatory-type diarrhea that is characterized by bloody, purulent, and mucoid stool [6]. This study also demonstrated that the WBC count and CRP level can be helpful in diagnosing appendicitis [22]. The sensitivity of combining both WBC counts and CRP levels was extremely high in children with acute appendicitis [23,24], while WBC counts or CRP alone did not aid in the diagnosis because normal values of both WBC and CRP were very rare in pediatric acute appendicitis. On the other hand, some studies have suggested that CRP may be more sensitive than WBC in detecting appendiceal perforation [25,26,27]. In fact, the diagnostic accuracy of WBC counts and CRP levels depends on the cutoff values and time from onset of symptoms to diagnosis [22]. It can explain our combined model has more diagnostic accuracy than the PAS score model.

This study has several limitations. First, this is a single-center study, and the identified predictors may not be able to be generalized to other institutions or countries. Second, the sample size was relatively small, and some potential predictors may not have been included in this study. Third, we did not perform model validation because of the inadequate sample sizes; therefore, further research is warranted for validation in order to accomplish better clinical application. 

In conclusion, this study is the first to identify the clinical predictors of acute appendicitis in children with gastroenteritis-like presentation. If a child initially presents with abdominal pain and symptoms mimicking gastroenteritis, accompanied with RLQ pain, an absence of stool OB, and higher WBC count and CRP level, a diagnosis of acute appendicitis would be more accurate. In our revised score system, a total of score >3 indicated a high probability of acute appendicitis. It may impact clinical care and substantially decrease the use of unnecessary CT scans in gastroenteritis patients. Our study assists clinical physicians in distinguishing acute appendicitis from acute gastroenteritis, enabling an early and precise diagnosis that can improve outcomes and may further prevent complications. 

## Figures and Tables

**Figure 1 children-07-00176-f001:**
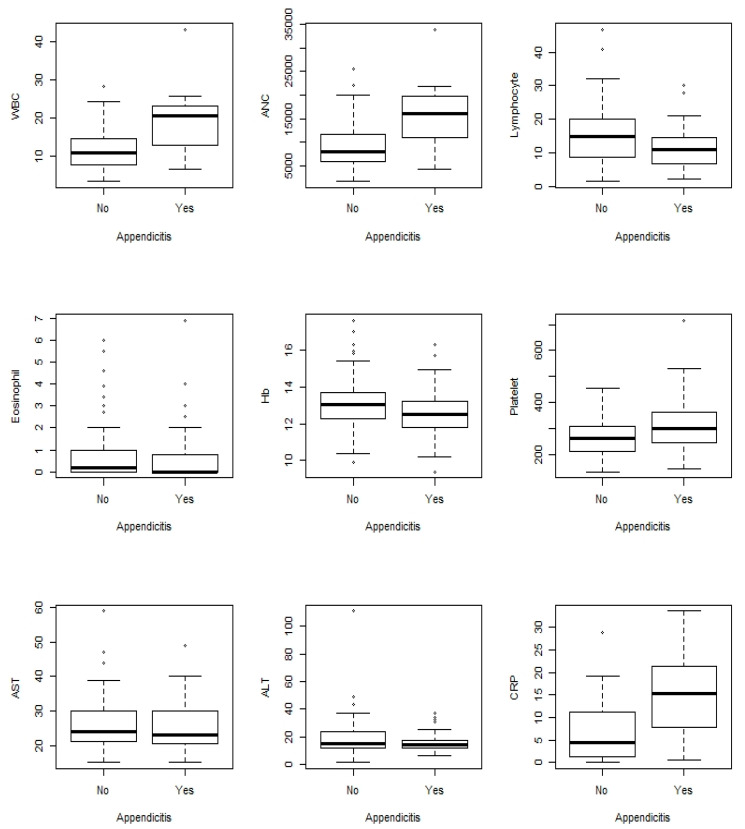
Boxplot for the laboratory data. WBC = white blood cell; ANC = absolute neutrophil count; AST =aspartate aminotransferase (U/L), ALT =alanine aminotransferase (U/L); CRP = C-reactive protein (mg/dL).

**Figure 2 children-07-00176-f002:**
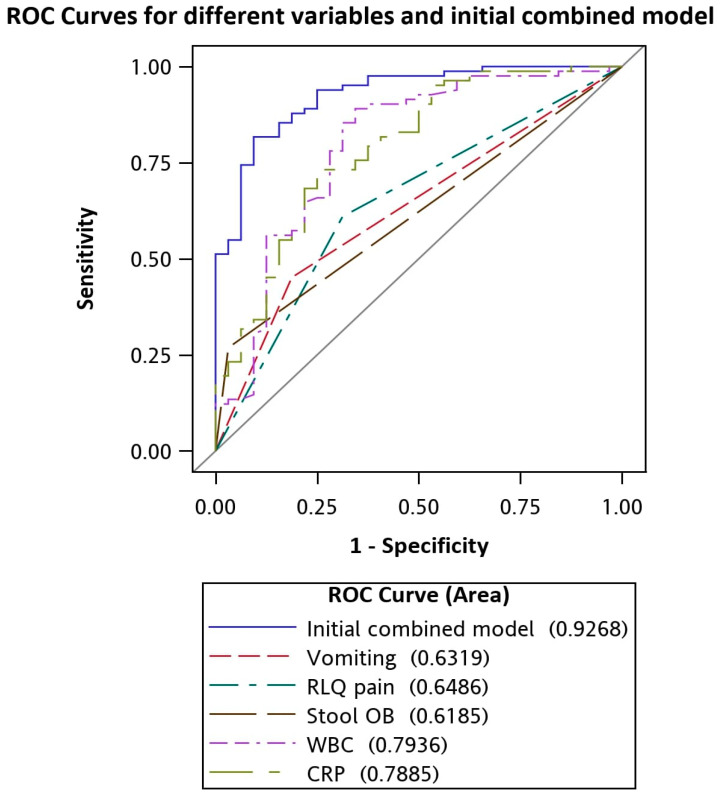
ROC curves for vomiting, RLQ pain, negative stool OB, CRP level, WBC level, and the initial combined model for differentiating between appendicitis and gastroenteritis.

**Figure 3 children-07-00176-f003:**
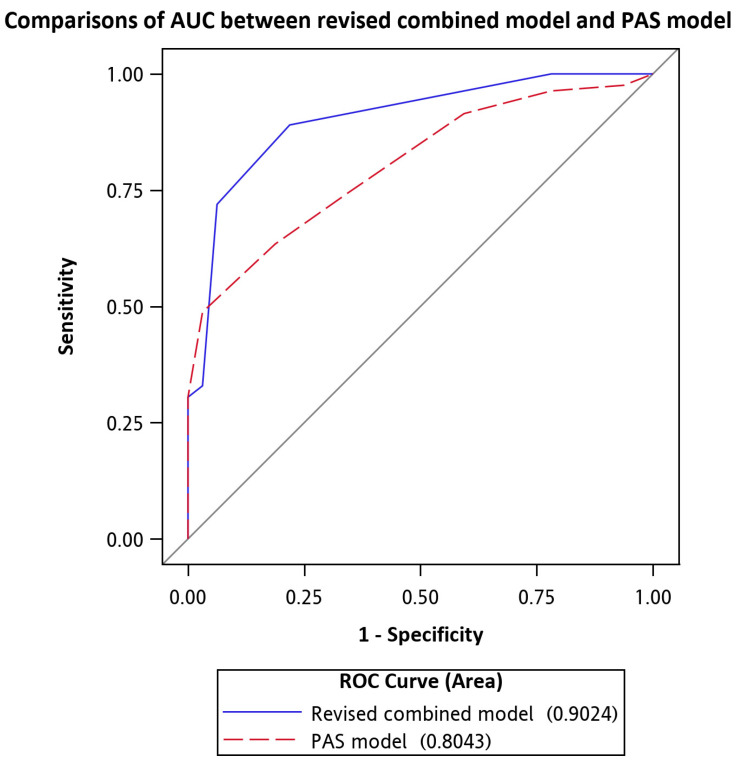
ROC curves for comparisons of the AUC between the revised combined model and the PAS model.

**Table 1 children-07-00176-t001:** Demographics, characteristics, and outcome in the appendicitis group and the enterocolitis group.

	Gastroenteritis Group (Group 0) N = 82	Appendicitis Group (Group 1) N = 32	*p*-Value
**Categorical characteristics**			
Sex (male)	50 (60%)	22 (69%)	0.577
Vomiting	45 (55%)	26 (81%)	0.010
Anorexia	78 (95%)	32 (100%)	0.575
Loose stool frequency			
1–3 times /day	36 (44%)	19 (59%)	0.163
>3 times /day	27 (33%)	5 (16%)	
Days with abdominal pain			
1–3 days	72 (88%)	15 (47%)	<0.001
>3 days	10 (12%)	17 (53%)	
Epigastric pain	14 (17%)	4 (13%)	0.776
Periumbilical pain	38 (46%)	11 (34%)	0.343
Diffuse pain	16 (20%)	7 (22%)	0.798
RLQ pain	32 (39%)	22 (69%)	0.008
Migration pain	15 (18%)	11 (34%)	0.112
Bowel sound			
hypoactive	7 (9%)	11 (34%)	0.002
normoactive	47 (57%)	15 (47%)	
hyperactive	28 (34%)	6 (19%)	
Peritoneal sign	13 (16%)	14 (44%)	0.004
Stool OB positive	22 (27%)	1 (3%)	0.004
Stool pus positive	16 (20%)	2 (6%)	0.094
Stool culture positive	28 (34%)	1 (3%)	<0.001
**Continuous characteristics**			
Age	10.46 ± 3.85	8.88 ± 4.51	0.059
Hospitalization days	5.06 ± 1.84	8.69 ± 4.08	<0.001
Fever duration (days)	1.87 ± 1.33	2.56 ± 2.33	0.267
Diarrhea days	1.76 ± 1.47	1.78 ± 2.11	0.52
PAS	4.93 ± 1.93	7.13 ± 1.58	<0.0001

Data are presented as mean value ± SD for continuous variables and the number and percentage of patients. RLQ pain = right lower quadrant; PAS = pediatric appendicitis score; OB = occult blood.

**Table 2 children-07-00176-t002:** The stool analysis between the two groups.

	Gastroenteritis Group (Group 0) N = 82	Appendicitis Group (Group 1) N = 32	*p*-Value
**Stool OB**			
negative	60	31	0.004
positive	22	1	
**Stool pus**			
negative	66	30	0.094
positive	16	2	

**Table 3 children-07-00176-t003:** Multivariable stepwise logistic regression of the laboratory model, clinical data model, and combined model.

Clinical Data Model Variable	β	SE	OR	95% CI	*p*-Value
Intercept	−2.01	0.84			0.017
Vomiting (Yes vs. No)	1.88	0.66	6.58	1.79–24.14	0.005
Days with abdominal pain (>3 days vs. others)	2.14	0.59	8.47	2.68–16.76	<0.001
RLQ pain (Yes vs. No)	1.73	0.60	5.63	1.72–18.41	0.004
Bowel sound (reference: hypoactive)					
Normoactive	−2.19	0.73	0.11	0.03–0.46	0.003
Hyperactive	−2.49	0.86	0.08	0.02–0.45	0.004
**Laboratory Model Variable**					
Intercept	−4.88	0.97			
WBC (per 10^3^/uL increase)	0.19	0.05	1.21	1.09–1.34	<0.001
CRP (per mg/dL increase)	0.14	0.04	1.15	1.07–1.24	<0.001
Stool OB (Yes vs. No)	−2.52	1.13	0.08	0.01–0.74	0.026
**Combined Model Variable**					
Intercept	−7.72	1.60			
Vomiting (Yes vs. No)	1.90	0.81	6.69	1.37–32.72	0.019
RLQ pain (Yes vs. No)	2.20	0.75	9.06	2.06–39.80	0.004
Stool OB (Yes vs. no)	−2.95	1.34	0.05	0.00–0.73	0.028
WBC (per 10^3^/uL increase)	0.18	0.06	1.20	1.07–1.34	0.002
CRP (per mg/dL increase)	0.17	0.05	1.19	1.09–1.30	<0.001

CI = confidence interval.

**Table 4 children-07-00176-t004:** Different variables and initial combined variables receiver operating characteristic (ROC) models associated with acute appendicitis.

ROC Model	Mann-Whitney		Sensitivity	Specificity	Cut-off Value	Assigned Points for Revised Combined Model
	AUC	95% CI				
Initial combined variables	0.93	0.88–0.98	0.82	0.91		
Vomiting	0.63	054–0.72	0.45	0.81		Yes:2; No:0
RLQ pain	0.65	0.55–0.75	0.61	0.69		Yes:2; No:0
Stool OB	0.62	0.56–0.68	0.27	0.97		Positive: −3; Negative:0
WBC	0.79	0.69–0.90	0.89	0.66	18.2	>18.2:1; ≦18.2:0
CRP	0.79	0.69–0.88	0.73	0.75	7.64	>7.64:1; ≦7.64:0

AUC= area under curve.

**Table 5 children-07-00176-t005:** Revised combined model and pediatric appendicitis score (PAS). ROC model associated with PAS model.

ROC Model	Mann–Whitney		Sensitivity	Specificity	Cut-off Model Score
	AUC	95% CI			
Revised combined model	0.90	0.84–0.96	0.89	0.78	3
PAS model	0.80	0.72–0.88	0.49	0.97	4
	**∆AUC**	**95% CI**	***p*-value**		
Revised combined model vs. PAS model	0.10	0.02–0.17	0.012		

**Table 6 children-07-00176-t006:** Revised model score.

Predictors	Point
**Vomiting**	
Yes	2
No	0
**RLQ pain**	
Yes	2
No	0
**Stool OB**	
Positive	−3
Negative	0
**WBC (10^3^/uL)**	
>18.20	1
≦18.20	0
**CRP (mg/dL)**	
>7.64	1
≦7.64	0

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
