# Peer review of "Making a Decision between Acute Appendicitis and Acute Gastroenteritis"

_children, 2020, doi:10.3390/children7100176_

Round 1
Reviewer 1 Report
This retrospective case-control study evaluated children who presented with abdominal pain and diarrhea and divided them into two groups; appendicitis (n=32) and enterocolitis (n=82). As children with appendicitis can present with diarrhea, the authors attempted to identify differentiating characteristic in these groups and identified 5 independent predictors of acute appendicitis (vomiting, RLQ pain, stool occult blood, WBC, and CRP).
Major Revisions:
- Enrollment criteria includes all those admitted with abdominal pain and diarrhea; and no exclusion. It seems unlikely that all children with abdominal pain and diarrhea had appendicitis or infectious enterocolitis. Were there no other diagnoses that were excluded?
- There is a decent number of patients that did not have data on the amount of diarrhea? How do we know that these were all patients that had diarrhea? How was diarrhea defined? Often, diarrhea is defined as >3 loose BM per day, but many these patients had 1-3 BM. This requires justification.
Minor Revisions:
- The PAS score is intended for patients who present with abdominal pain- but this scoring symptoms is applied to patients who present with diarrhea. What proportion of patients with appendicitis have diarrhea? This will help with the importance of the study. It also seems that this would not replace the PAS but would supplement it in patients who present with diarrheal symptoms.
- How was infectious enterocolitis defined- based on provider dx?
- Did all of the patients have to have appendicitis on pathology to qualify for group 1?
- If called group 0 and group 1 in the text- this should also be included in the tables.
- Figure 1 is not needed- same data presented in table 2.
- How many patients were perforated at time of hospital evaluation? 42% is mentioned in the discussion but not in the results.
- The authors conclude that their study can help “enabling early precise diagnosis that can improve outcomes and prevent complications of appendiceal perforation.” However, the study notes that “up to 42% of patients had appendiceal perforation.” Therefore, the study can conclude that it can assist in differentiating enterocolitis from appendicitis, but not in preventing perforation as that was not the aim of the study.
- Need more discuss on what is in the PAS and how this scoring system is different.
Author Response
Dear reviewer:
We sincerely appreciate all the precious comments. Those are so valuable recommendations that help us to modify and improve the quality of study. We revised the manuscript in accordance with the comments, and carefully proof-read the manuscript to minimize typographical, grammatical, and bibliographical errors. The questions were answered below. Should you have any questions, please contact us without hesitate.
Major Revisions
- Enrollment criteria includes all those admitted with abdominal pain and diarrhea; and no exclusion. It seems unlikely that all children with abdominal pain and diarrhea had appendicitis or infectious enterocolitis. Were there no other diagnoses that were excluded?
Reply: Thanks for your suggestions for the enrollment criteria. We include patients who had abdominal pain with diarrhea and/or vomiting, and final diagnosis with appendicitis or gastroenteritis. Thus, other diagnoses were excluded. We will also change the term of “infectious enterocolitis” to “acute gastroenteritis”, in order to make the definition more precise.
- There is a decent number of patients that did not have data on the amount of diarrhea? How do we know that these were all patients that had diarrhea? How was diarrhea defined? Often, diarrhea is defined as >3 loose BM per day, but many these patients had 1-3 BM. This requires justification.
Reply: Thanks for the suggestions. We enrolled patients who had abdominal pain with diarrhea and/or vomiting, therefore, not all patients had diarrhea. The most common manifestations of acute gastroenteritis are diarrhea and/or vomiting. Our definition of diarrhea are the passage of loose or watery stools and increased frequency of defecation. The more precise definition of diarrhea is stool volume more than 10 g/kg/day, but it is not practical in clinical application. Although the World Health Organization (WHO) defines diarrhea as the passage of three or more loose or watery stools per day, absolute limits of normalcy are difficult to define. Regardless of the actual number of stools or their water content, the deviation of usual stool pattern, such as loose, bloody, purulent, or mucoid stool, may be more appliable.
Minor Revisions:
- The PAS score is intended for patients who present with abdominal pain- but this scoring symptoms is applied to patients who present with diarrhea. What proportion of patients with appendicitis have diarrhea? This will help with the importance of the study. It also seems that this would not replace the PAS but would supplement it in patients who present with diarrheal symptoms.
Reply: In our study, proportion of patients with appendicitis have diarrhea is 75%. Our scoring symptoms is applied to patient who had symptoms of gastroenteritis, including abdominal with diarrhea and/or vomiting. It indeed would not replace the PAS but would supplement it in patients with appendicitis mimicking gastroenteritis.
- How was infectious enterocolitis defined- based on provider dx?
Reply: The definition of enterocolitis is based on clinical symptoms such as fever with diarrhea and/or vomiting and final stool culture result as well.
- Did all of the patients have to have appendicitis on pathology to qualify for group 1?
Reply: There were 84% patients in group 1 having pathology report, and the rest of 16% patients were diagnosed based on abdominal CT report.
- If called group 0 and group 1 in the text- this should also be included in the tables.
Reply: We will make the adjustment of the tables and will add group 0 and group 1 into the tables.
- Figure 1 is not needed- same data presented in table 2.
Reply: We will make the adjustment of the table and delete the same data presented in the table.
- How many patients were perforated at time of hospital evaluation? 42% is mentioned in the discussion but not in the results.
Reply: The rate of appendiceal perforation with abscess formation in our study was 42%. We will add the description into our result.
- The authors conclude that their study can help “enabling early precise diagnosis that can improve outcomes and prevent complications of appendiceal perforation.” However, the study notes that “up to 42% of patients had appendiceal perforation.” Therefore, the study can conclude that it can assist in differentiating enterocolitis from appendicitis, but not in preventing perforation as that was not the aim of the study.
Reply: Indeed, the perforation rate in out study is up to 42%. Our study can assist in differentiating acute gastroenteritis from appendicitis, but the whether it can prevent perforation may questionable.
- Need more discuss on what is in the PAS and how this scoring system is different.
Reply: We will add more discussion on PAS. PAS is a weighted clinical scoring system with eight clinical features for accessing patients with abdominal pain and diagnosing appendicitis in children. The score combines history, physical, and laboratory data to assist in the diagnosis. Migration of pain, anorexia, vomiting, fever >38°C, leukocytosis and polymorphonuclear neutrophilia were assigned a score of 1. RLQ pain and cough/percussion/hopping tenderness were assigned a score of 2. A score of >6 is compatible with the diagnosis of appendicitis in this previous study. In our study, we enrolled patients with abdominal pain and symptoms mimicking gastroenteritis. Multiple logistic regression analysis yielded a model comprising 5 variables, and we gave each predictor an assigned point value to calculate the revised combined model score. Our score would be more helpful for assessing cases with appendicitis mimicking acute gastroenteritis in children.
Thank you for all the kind advice.
Please see the attachment.

Reviewer 2 Report
This is an interesting study by Lu et al regarding the development of a model for the early diagnosis of acute appendicitis vs infectious enterocolitis in children admitted with abdominal pain and diarrhea. This is a valid clinical problem since delays in the recognition of appendicitis in these patients may lead to increased complications and morbidity. The researchers compare their model with the pediatric appendicitis score (PAS) and conclude that, in this subset of patients, their model is superior, but they come to this conclusion based only on the training data.
The study is interesting, although not particularly novel, and could be used as a base for further research in this field.
However, there are several issues that need to be addressed by the authors
Major points
1. The authors did not use any form of correction for multiple comparisons in the initial analysis of the independent variables they studied in a total of 114 patients. This could mean that some of the significant results presented in table 1 might be due to statistical noise.
2. Not all continuous variables presented in Table 1 seem to have normal distribution based on the boxplots in Figure 1. This suggests that the t-test might have not been the proper test for these variables.
3. There are discrepancies between the text in section 3.2 and table 3 (e.g. results presented for CRP are OR 1,19 with CI 1,19-1,30 in the text and 1,15 with CI 1,07-1,24 in the table).
4. As indicated in the limitations of the study, it is small and single-center, the model has not been validated and the results could be the result of local characteristics. Furthermore, having an AUC analysis, although useful, it is not helpful for clinicians. The sensitivity and specificity presented in table 5 should be explained based on clear cut-offs. If a PAS >6 was used for this table, a clear cut-off should be presented for the author’s combined model to explain the result. It would also be useful to present the model in a way that can be used by other researchers e.g. by converting it to a score.
Minor points
1. Some language editing and copy-editing are needed in the text.
2. Table 3 line 5. “abdominal days” should be “days with abdominal pain”
Author Response
Dear reviewer:
We sincerely appreciate all the precious comments. Those are so valuable recommendations that help us to modify and improve the quality of study. We revised the manuscript in accordance with the comments, and carefully proof-read the manuscript to minimize typographical, grammatical, and bibliographical errors. The questions were answered below. Should you have any questions, please contact us without hesitate.
Major points
- The authors did not use any form of correction for multiple comparisons in the initial analysis of the independent variables they studied in a total of 114 patients. This could mean that some of the significant results presented in table 1 might be due to statistical noise.
Reply: Thanks for your recommendation about adjustment for multiple testing in our study. We would adopt Benjamini-Hochberg method to control the false discovery rate of multiple testing in our study. In our study, the false discovery rate should be less than 0.0167 to achieve significant difference.
Reference:
Benjamini, Y., & Hochberg, Y. (1995). Controlling the false discovery rate: a practical and powerful approach to multiple testing. Journal of the Royal statistical society: series B (Methodological), 57(1), 289-300.
- Not all continuous variables presented in Table 1 seem to have normal distribution based on the boxplots in Figure 1. This suggests that the t-test might have not been the proper test for these variables.
Reply: Thanks for your suggestion. We would change to Mann–Whitney U test for comparisons of continuous variables.
- There are discrepancies between the text in section 3.2 and table 3 (e.g. results presented for CRP are OR 1,19 with CI 1,19-1,30 in the text and 1,15 with CI 1,07-1,24 in the table).
Reply: In the table 3, we performed multivariable stepwise logistic regression for clinical variables and laboratory variables to develop a clinical predictive model and laboratory predictive model separately. Then we analyzed clinical variables and laboratory variables together to develop a combined mode. Thus, the OR and CI value of CRP are different between the two columns.
- As indicated in the limitations of the study, it is small and single-center, the model has not been validated and the results could be the result of local characteristics. Furthermore, having an AUC analysis, although useful, it is not helpful for clinicians. The sensitivity and specificity presented in table 5 should be explained based on clear cut-offs. If a PAS >6 was used for this table, a clear cut-off should be presented for the author’s combined model to explain the result. It would also be useful to present the model in a way that can be used by other researchers e.g. by converting it to a score.
Reply: Thanks for the suggestions. Indeed, the model has not been validated for general application and could be the result of local characteristics; therefore, further study is warranted. We gave each predictor an assigned point value to calculate the revised combined model score. A score of >3 in our revised model indicated a high probability of acute appendicitis. The score may be more applicable for clinical practice.
Minor points
1. Some language editing and copy-editing are needed in the text.
Reply: We will make the language editing and copy-editing in the text.
2. Table 3 line 5. “abdominal days” should be “days with abdominal pain”
Reply: We will make the adjustment from “abdominal days” to days with abdominal pain.”
Thank you for the kind advice.
Please see the attachment.

Round 2
Reviewer 2 Report
The authors have properly responded to all of my previous points. The revised version is very improved.
Minor points
- The authors should provide the FDR chosen for the Benjamini-Hochberg
- Some of the data provided in the original version in table 1 (the last few rows) have been deleted. Is this a mistake? If not please provide these rows since they are helpful for the reader.
- Please correct the phrase in lines 385-387.
Author Response
Dear reviewer:
We sincerely appreciate the precious comments. The questions were answered below.
Should you have any questions, please contact us without hesitate.
Minor revisions:
- The authors should provide the FDR chosen for the Benjamini-Hochberg
Reply: Thank you for the suggestion. The FDR chosen for B-H method is 5% in our study. The false discovery rate should be less than 0.0167 to achieve significant difference.
- Some of the data provided in the original version in table 1 (the last few rows) have been deleted. Is this a mistake? If not please provide these rows since they are helpful for the reader.
Reply: Thank you for the suggestion. There is some of the same data among table 1 and figure1; therefore, the repeated lab data provided in the original version in table 1 was deleted. We will provide the deleted lab data in Supplementary Table 1.
- Please correct the phrase in lines 385-387.
Reply: Thank you for the suggestion. We are sorry that there are no lines 385-387 in the manuscript.
Thank you for all the kind advice.
Please see the attachment
